# ADAPT ON-THE-GO: BEHAVIOR MODULATION FOR SINGLE-LIFE ROBOT DEPLOYMENT

## ABSTRACT

To succeed in the real world, robots must cope with situations that differ from those seen during training. We study the problem of adapting on-the-fly to such novel scenarios during deployment, by drawing upon a diverse repertoire of previously-learned behaviors. Our approach, RObust Autonomous Modulation (ROAM), introduces a mechanism based on the perceived value of pre-trained behaviors to select and adapt pre-trained behaviors to the situation at hand. Crucially, this adaptation process all happens within a single episode at test time, without any human supervision. We provide theoretical analysis of our selection mechanism and demonstrate that ROAM enables a robot to adapt rapidly to changes in dynamics both in simulation and on a real Go1 quadruped, even successfully moving forward with roller skates on its feet. Our approach adapts over 2x as efficiently compared to existing methods when facing a variety of out-of-distribution situations during deployment by effectively choosing and adapting relevant behaviors on-the-fly.

Figure 1: **On-The-Go Adaptation via Robust Autonomous Modulation (ROAM).** An agent will inevitably encounter a wide variety of situations during deployment, and handling such situations may require a variety of different behaviors. We propose Robust Autonomous Modulation (ROAM), which dynamically employs and adapts relevant behaviors as different situations arise during a single trial of deployment.

## 1 INTRODUCTION

A major obstacle to the broad application of sequential decision-making agents is their inability to adapt to unexpected circumstances, which limits their uses largely to tightly controlled environments. Even equipped with prior experience and pre-training, agents will inevitably encounter out-of-distribution (OOD) situations at deployment time that may require a large amount of on-the-fly adaptation. In this work, we aim to enable a robot to autonomously handle novel scenarios encountered during deployment, while drawing upon a diverse set of pre-trained behaviors that may improve its versatility. We hypothesize that by doing some of the adaptation in the space of pre-trained behaviors (rather than only in the space of parameters), we can much more quickly react and adapt to novel circumstances. Accumulating a set of different behavior policies is a relatively straightforward

task through online or offline episodic reinforcement learning, using different reward functions or skill discovery methods. However, the on-the-fly selection and adaptation of these behaviors during deployment, particularly in novel environments, present significant challenges. Consider tasking a quadrupedal robot that has acquired many different behaviors, e.g., walking, crouching, and limping via training in simulation, with a search-and-rescue mission in the real world. When deployed on this task with unstructured obstacles, the robot may bump into an obstacle and damage its leg, and it must be able to dynamically adapt its choice of behaviors to continue its mission with the damage.

Existing adaptation methods often operate within the standard, episodic training paradigm where the agent is assumed to be reset and have another chance to attempt the task each time (Cully et al., 2015; Song et al., 2020; Julian et al., 2020). However, these are idealized conditions that in practice often rely on human intervention. In the above search-and-rescue example, the robot's state cannot be arbitrarily restored; during deployment, the robot cannot be repaired, and it is not feasible for a human to fetch it if it gets stuck in a situation it is not equipped to handle. This situation necessitates adapting at test time both quickly and autonomously, to succeed at the task within a single episode. Therefore, we frame our problem setting as an instantiation of single-life deployment (Chen et al., 2022), where the agent is given prior knowledge from 'past lives' but evaluated on its ability to successfully complete a given task later in a 'single-life' trial, during which there are no human-provided resets. The robot is provided with a diverse set of prior behaviors trained through episodic RL, and the single life poses a sequence of new situations.

To solve this problem, the robot must identify during deployment which behaviors are most suited to its situation at a given timestep and have the ability to fine-tune those behaviors in real time, as the pre-trained behaviors may not optimally accommodate new challenges. We introduce a simple method called RObust Autonomous Modulation (ROAM), which foremost aims to quickly identify the most appropriate behavior from its pre-trained set at each point in time during single-life deployment. Rather than introducing an additional component like a high-level controller to select behaviors, we leverage the value function of each behavior. The value function already estimates whether each policy will be successful, but may not be accurate for states that were not encountered during training of that behavior. Therefore, prior to deployment, we fine-tune each behavior's value function with a regularized objective that encourages behavior identifiability: the regularizer is a behavior classification loss. Then, at each step during deployment, ROAM samples a behavior proportional to its classification probability, executes an action from that behavior, and optionally fine-tunes the selected behavior for additional adaptation.

The main contribution of this paper is a simple algorithm for autonomous, deployment-time adaptation to novel scenarios. In our theoretical analysis, we show that at a given state, with the additional cross-entropy regularizer, ROAM can constrain each behavior's value to be lower than the value of behaviors for which that state appears more frequently. Consequently, our method incentivizes each behaviors to differentiate between familiar and unfamiliar states, allowing ROAM to better recognize when a behavior will be useful. We conduct experiments on both simulated locomotion tasks and on a real Go1 quadruped robot. In simulation, our method completes the deployment task more than two times faster on average than existing methods, including two prior methods designed for fast adaptation. We also empirically analyze how the additional cross-entropy term in the loss function of ROAM contributes to more successful utilization of the prior behaviors. Furthermore, ROAM enables the Go1 robot to adapt on-the-go to various OOD situations without human interventions or supervision in the real world. With ROAM, the robot can successfully pull heavy luggage, pull loads with dynamic weights, and even slide forward with two roller skates on its front feet, even though it never encountered loads or wore roller skates during training.

## 2 RELATED WORK

We consider the problem of enabling an agent to act robustly when transferring to unstructured test-time conditions that are unknown at train-time. One common instantiation of this problem is transfer to different dynamics, e.g., in order to transfer policies trained in simulation to the real world. A popular approach in achieving transfer under dynamics shift is domain randomization, i.e., randomizing the dynamics during training (Cutler et al., 2014; Rajeswaran et al., 2016; Sadeghi & Levine, 2016; Tobin et al., 2017; Peng et al., 2018; Tan et al., 2018; Yu et al., 2019; Akkaya et al., 2019; Xie et al., 2021; Margolis et al., 2022; Haarnoja et al., 2023) to learn a robust policy. Our approach is similar in that it takes advantage of different MDPs during training; however, a key component of ROAM is to leverage and modulate diverse skills rather than a single, robust policy.

We find in Section 5 that challenging test-time scenarios may require distinctly different behaviors at different times, and we design our method to be robust to those heterogeneous conditions.

A class of methods that use domain randomization has also utilized different 'behaviors' instead of a one-size-fits-all policy. Especially effective in locomotion applications, these methods involve training policies that exhibit different behavior when conditioned on dynamics parameters, then distilling these policies into one that can be deployed in target domains where this information is not directly observable. The train-time supervision can come in the form of the parameter values (Yu et al., 2017; Ji et al., 2022) or a learned representation of them (Lee et al., 2020; Kumar et al., 2021). Thereafter, there are several ways prior work have explored utilizing online data to identify which behavior is appropriate on-the-fly, e.g., using search in latent space (Yu et al., 2019; Peng et al., 2020; Yu et al., 2020b), or direct inference using proprioceptive history (Lee et al., 2020; Kumar et al., 2021; Fu et al., 2022), or prediction based on egocentric depth (Miki et al., 2022; Agarwal et al., 2022; Zhuang et al., 2023; Yang et al., 2023). In this work, we do not rely on domain-specific information nor external supervision for when particular pre-trained behaviors are useful. Moreover, in contrast to many of the above works, we focus on solving tasks that may be OOD for all prior behaviors individually. By re-evaluating and potentially switching behaviors at every timestep, ROAM can take the most useful parts of different behaviors to solve such tasks, as we find in Section 5.

Meta-RL is another line of work that achieves rapid adaptation without privileged information by optimizing the adaptation procedure during training (Wang et al., 2016; Duan et al., 2016; Finn et al., 2017; Nagabandi et al., 2018; Houthooft et al., 2018; Rothfuss et al., 2018; Rusu et al., 2018; Mendonca et al., 2020; Song et al., 2020) to be able to adapt quickly to a new situation at test-time. Meta-RL and the aforementioned domain randomization-based approaches entangle the training processes with the data collection, requiring a lot of *online* samples that are collected in a particular way for pre-training. The key conceptual difference in our approach is that ROAM is *agnostic to how the pre-trained policies and value functions are obtained*. Moreover, while meta-RL methods often use hundreds of pre-training tasks, or more, our approach can provide improvements in new situations even with a relatively small set of pre-trained behaviors (e.g. just 4 different behaviors improve performance in Section 5). Other transfer learning approaches adapt the weights of the policy to a new environment or task, either through rapid zero-shot adaptation (Hansen et al., 2020; Yoneda et al., 2021; Chen et al., 2022) or through extended episodic online training (Khetarpal et al., 2020; Rusu et al., 2016; Eysenbach et al., 2020; Xie et al., 2020; Xie & Finn, 2021). Unlike these works, we focus on adaptation within a single episode to a variety of different situations.

Another rich body of work considers how to combine prior behaviors to solve long-horizon tasks, and some of these works also focus on learning or discovering useful skills (Gregor et al., 2016; Achiam et al., 2018; Eysenbach et al., 2018; Nachum et al., 2018a; Sharma et al., 2019; Baumli et al., 2021; Laskin et al., 2022; Park & Levine, 2023). Many of these methods involve training a high-level policy that learns to compose learned skills into long-horizon behaviors (Bacon et al., 2017; Peng et al., 2019; Lee et al., 2019; Sharma et al., 2020; Strudel et al., 2020; Nachum et al., 2018b; Chitnis et al., 2020; Pertsch et al., 2021; Dalal et al., 2021; Nasiriany et al., 2022). We show in Section 5 that training such a high-level policy is not needed for effective on-the-go behavior selection. Moreover, our work does not focus on where the behaviors come from – they could be produced by these prior methods, or their rewards could be specified manually. Instead, we focus on quickly selecting and adapting the most suitable skill in an OOD scenario, without requiring an additional online training phase to learn a hierarchical controller.

## 3 PRELIMINARIES

In this section, we describe some preliminaries and formalize our problem statement. We are given a set of $n$ prior behaviors, where each behavior $i$ is trained through episodic RL for a particular MDP $\mathcal{M}_i = (\mathcal{S}, \mathcal{A}, \mathcal{P}_i, \mathcal{R}_i, \rho_0, \gamma)$ where $\mathcal{S}$ is the state space, $\mathcal{A}$ is the agent's action space, $\mathcal{P}_i(s_{t+1}|s_t, a_t)$ represents the environment's transition dynamics, $\mathcal{R}_i : \mathcal{S} \rightarrow \mathbb{R}$ indicates the reward function, $\rho_0 : \mathcal{S} \rightarrow \mathbb{R}$ denotes the initial state distribution, and $\gamma \in [0, 1)$ denotes the discount factor. Each of the $n$ MDPs $\mathcal{M}_i$ has potentially different dynamics and reward functions $\mathcal{P}_i$ and $\mathcal{R}_i$, often leading to different state visitation distributions. Each behavior corresponds to a policy $\pi_i$ and a value function $V_i$ as well as a buffer of trajectories $\tau \in \mathcal{D}_i$ collected during this prior training and relabeled with the reward $\mathcal{R}_{\text{target}}$ from the target MDP. At test time, the agent interacts with a target MDP defined by $\mathcal{M}_{\text{target}} = (\mathcal{S}, \mathcal{A}, \mathcal{P}_{\text{target}}, \mathcal{R}_{\text{target}}, \rho_0, \gamma)$, which presents an aspect of novelty not present in any of the prior MDPs, in the form of new dynamics $\mathcal{P}_{\text{target}}(s_{t+1} \mid s_t, a_t)$, which may *change over the course*

*of the test-time trial.* We operate in a single-life deployment setting (Chen et al., 2022) that aims to maximize $J = \sum_{t=0}^{h} \gamma^t \mathcal{R}(s_t)$, where $h$ is the trial horizon, which may be $\infty$. The agent needs to complete the desired task in this target MDP in a single life without any additional supervision or human intervention by effectively selecting and adapting the prior behaviors to the situation at hand.

Off-policy reinforcement learning (RL) algorithms train a parametric Q-function, represented as $Q_\theta(s, a)$, via iterative applications of the Bellman optimality operator, expressed as

$$B^* Q(s, a) = r(s, a) + \gamma \mathbb{E}_{s' \sim P(s'|s,a)} \left[ \max_{a'} Q(s', a') \right].$$

In actor-critic frameworks, a separate policy is trained to maximize Q-values. These algorithms alternate between policy evaluation, which involves the Bellman operator $B^\pi Q = r + \gamma P^\pi Q$, and policy improvement, where the policy $\pi(a|s)$ is updated to maximize expected Q-values.

The dataset $D_\beta$, containing tuples $(s, a, r, s')$ gathered by a behavior policy $\pi_\beta$, typically lacks full coverage of all possible transitions. Therefore, an empirical Bellman operator, denoted as $\hat{B}^\pi$, is used during policy evaluation. Specifically, the respective policy evaluation and improvement updates are:

$$Q^{k+1} \leftarrow \arg\min_Q \mathbb{E}_{s,a,s' \sim D} \left[ \left( r(s, a) + \gamma \mathbb{E}_{a' \sim \pi^k(a'|s')}[Q^k(s', a')] - Q(s, a) \right)^2 \right]$$

$$\pi^{k+1} \leftarrow \arg\max_\pi \mathbb{E}_{s \sim D, a \sim \pi^k(a|s)} \left[ Q^{k+1}(s, a) \right].$$

We use a state-of-the-art off-policy actor-critic algorithm RLPD (Ball et al., 2023) as our base algorithm for pre-training and fine-tuning, as it has been shown to be effective especially in the latter regime. RLPD builds on regularized soft actor-critic (SAC) (Haarnoja et al., 2018), using Layer Normalization (Ba et al., 2016) in particular as a key component.

## 4 ROBUST AUTONOMOUS MODULATION

We now present our method, Robust Autonomous Modulation (ROAM), which fine-tunes value functions with an additional loss and provides a mechanism for choosing among them, so that at deployment time, the agent can quickly react to its current situation *at every timestep* by honing in on the most appropriate behavior from its prior behaviors. Our key observation is that with proper regularization, value functions provide a good indication of how well different behaviors will perform, so we can leverage them to quickly identify appropriate behaviors in a given situation, which circumvents the need to learn a separate meta-controller or adaptation module. In the following subsections, we describe our method in detail and provide theoretical analysis on how our method leads to more efficient utilization of the prior behaviors by encouraging the value functions of the behaviors to better distinguish between familiar and OOD states.

### 4.1 ALGORITHM DESCRIPTION

**Behavior Modulation using Value Functions.** The core idea of our method is to directly utilize the expressive power of value functions for adaptive behavior selection, as they inherently contain detailed information about potential rewards associated with different states for each behavior. We propose to use value functions of the behaviors to select the most appropriate behavior for a given state—namely, at each timestep during deployment, choosing one of the behaviors with the high values at that state. Since we already have access to value functions from pre-training the behaviors, this approach does not require any additional training or data collection. Using value functions as the proxy selector also gives much more versatility to the selection mechanism, which can be flexibly controlled on the go by updating the value functions of different behaviors.

However, naively using the pre-trained value functions may not lead to high-reward behaviors, due to overestimation of the value functions on new states, as the pre-trained Q-functions may not generalize well to OOD situations. Recent studies in offline RL, for example, have observed an overestimation bias (Levine et al., 2020) due to OOD training-time actions, which can lead to poor performance when deploying the policy in a new environment. To mitigate these issues, works in offline RL have proposed a number of various modifications aimed at regularizing either the policy or the value function (Kumar et al., 2020; Yu et al., 2020a; 2021). Although our setting is different, as we deal with OOD states, we face a similar problem of poor generalization of the value functions of the prior behaviors to unfamiliar situations. In the following section, we describe how we can conservatively regularize the value functions to improve their generalization.

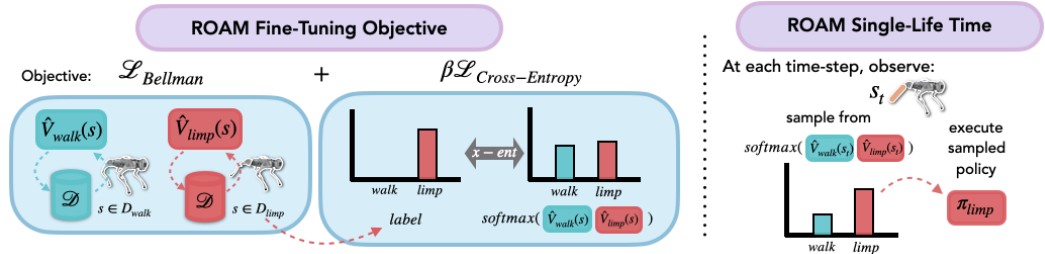

Figure 2: **Robust Autonomous Modulation (ROAM).** During the initial fine-tuning phase of ROAM, we fine-tune each behavior using its existing data buffer and standard Bellman error, with an additional cross-entropy loss between the softmax values of all behaviors and the behavior index (as the label) from which the state was visited. Then at test-time, at each time-step, we sample from the softmax distribution of the behaviors' values at the current state and execute the sampled policy.

**Fine-Tuning Value Functions with ROAM.** We desire value functions that accurately reflect the expected reward of using a behavior at a given state. Thus, we would like to fine-tune the value functions of the behaviors to minimize overestimation of the values at states for which a different behavior is more suitable. We can do so by incentivizing the value functions of the behaviors to be higher for states that are visited by that behavior and lower for states visited by other behaviors.

Given a set of prior behaviors $\mathcal{B}$ with policies $\pi_i$ and critics $Q_i$, we first fine-tune the value functions of the behaviors with an additional cross-entropy loss on top of the Bellman error that takes in the values at a given state as the logits. Values at a given state $s$ for behavior $i$ are obtained by averaging $Q_i(s, a)$ over $N = 5$ sampled actions $a \sim \pi_i(\cdot \mid s)$. More formally, with each prior data buffer $\mathcal{D}_i$, we fine-tune the critic $Q_i(s, a)$ of each behavior $i$ with the following update:

$$
\begin{aligned}
\mathcal{L}_{\text{fine-tune}} &= (1 - \beta)\mathcal{L}_{\text{Bellman}} + \beta\mathcal{L}_{\text{cross-entropy}} \\
&= (1 - \beta) \sum_i \mathbb{E}_{s,a,s' \sim \mathcal{D}_i} \left[ \left( r(s, a) + \gamma \mathbb{E}_{a' \sim \pi_i(a'|s')} Q_i(s', a') - Q_i(s, a) \right)^2 \right] \\
&\quad + \beta \sum_j \mathbb{E}_{s \sim \mathcal{D}_j} \left[ -\log \frac{\exp(V_j(s))}{\sum_{k=1}^n \exp(V_k(s))} \right],
\end{aligned}
\tag{1}
$$

where $0 < \beta < 1$ is a hyperparameter, $V_i(s) = \mathbb{E}_{a \sim \pi_i(a|s)}[Q_i(s, a)]$ is the average value of behavior $i$ at state $s$, and $\mathcal{D}_i$ is a replay buffer collected by behavior $i$. Consider the derivative of the cross-entropy term with respect to the value functions $V_i(s)$ and $V_k(s)$, where $i \neq k$, for a state $s$ visited by behavior $i$:

$$
\frac{\partial \mathcal{L}_{\text{cross-entropy}}}{\partial V_i(s)} = \frac{\exp(V_i(s))}{\sum_{j=1}^n \exp(V_j(s))} - 1 < 0, \quad \frac{\partial \mathcal{L}_{\text{cross-entropy}}}{\partial V_k(s)} = \frac{\exp(V_k(s))}{\sum_{j=1}^n \exp(V_j(s))} > 0.
$$

So when minimizing the cross-entropy loss, the value function $V_i(s)$ will be pushed up (since its derivative is negative), and $V_k(s)$ for $k \neq i$ will be pushed down. Thus, the cross-entropy loss term in Equation 1 pushes up the value functions of the behaviors for states that are visited by that behavior and pushes down for states that are visited by other behaviors. The value functions are then less likely to overestimate at OOD states, enabling the behaviors to specialize in different parts of the state space, which will help us at test time to better infer an appropriate behavior from the current state.

**Full Procedure and Single-Life Deployment.** To summarize the full procedure of ROAM, we are given a set of policies $\pi_i$ and critics $Q_i$, and a set of prior data buffers $\mathcal{D}_i$ for each behavior. Alternatively, this can be relaxed and we can assume that we are given a set of prior data buffers $\mathcal{D}_i$ for each behavior, and we can train the policies $\pi_i$ and critics $Q_i$ using these buffers with offline RL. We then fine-tune the value functions of the behaviors with the additional cross-entropy loss term in Equation 1 to obtain the final value functions $V_i$.

Then during each timestep at test time, we sample a behavior from the softmax distribution given by the behaviors' values $V_i$ at the current state. Formally, given the current state $s$, we sample an action $a_i \sim \pi_i(a|s)$ from behavior $i$ with probability proportional to $\exp(V_i(s))$. The transition $(s_t, a_t, r_t, s_{t+1})$ is then added to the online buffer $\mathcal{D}_{\text{online}}^i$ for behavior $i$ and the critic $V_i$ and policy

$\pi_i$ are fine-tuned using data from $\mathcal{D}^i_{\text{online}}$. In this manner, we can choose and adapt the most suitable behavior for a given state on-the-fly. The ROAM fine-tuning objective and single-life deployment are depicted in Figure 2 and the full algorithm is summarized in Algorithms 1 and 2 in Appendix A.2.

## 4.2 THEORETICAL ANALYSIS

Next, we theoretically analyze ROAM to show that the additional cross-entropy loss in ROAM will lead to more suitable behaviors being chosen at each timestep. In particular, ROAM rescales the value functions of the behaviors so that they are less likely to overestimate in states that are out of distribution for that behavior. Our main result, given in Theorem 4.2, is that with ROAM, for some weight $\beta > 0$ on the cross-entropy term, at a given state, ROAM constrains each behavior's value to be lower than the value of behaviors for which that state appears more frequently. This theorem gives us *conservative generalization* by reducing value overestimation in unfamiliar states–specifically, if at least one behavior is familiar with the current state, our chosen behavior will not have much worse performance than its value function estimate. Full proofs of the statements in this section are presented in Appendix A.1.

Notationally, let behavior $i$ be associated with a reward $R_i(s) = \mathbb{E}_{a \sim \pi_i(\cdot|s)}[R_i(s,a)]$ and dynamics function $P_i$ for policy $\pi_i$. Our modified Bellman operator is $\hat{B}^\pi V = \left[\hat{B}^{\pi_i} V_i\right]_{i=1}^n$, where for each value function $V_i$, for $\beta \in (0,1)$,

$$(\hat{B}^{\pi_i} V_i)(s) = (1-\beta)\left(R_i(s) + \gamma \sum_{s' \in S, a \in A} \pi_i(a|s) P_i(s'|s,a) V_i(s')\right) + \beta\left(\frac{\exp(\tau V_i(s))}{\sum_{k=1}^n \exp(\tau V_k(s))}\right).$$

**Lemma 4.1.** *There exists a temperature $\tau > 0$ for which our modified Bellman operator $\hat{B}^\pi V = \left[\hat{B}^{\pi_i} V_i\right]_{i=1}^n$ is a contraction under the $\mathcal{L}^\infty$ norm.*

By Lemma 4.1, for some temperature $\tau > 0$, and by the contraction mapping theorem, our modified Bellman operator will converge to a fixed point. In the following theorem, we characterize this fixed point and use it to analyze how ROAM will adjust value estimates based on degree of familiarity. Let $p_i(s)$ denote the state visitation probability for a behavior $i$ at state $s$.

**Theorem 4.2.** *For any state $s$ that is out of distribution for behavior $i$ and is in distribution for another behavior $j$, i.e. $p_i(s) \ll p_j(s)$, if $0 < \beta < 1$ is chosen to be large enough, then the value of behavior $i$ learned by ROAM will be bounded above compared to value of behavior $j$, i.e., $V_i(s) \leq V_j(s)$.*

As a result, for any states $s$ that are out of distribution for behavior $i$, if we choose $\beta$ large enough, the value learned $V_i(s)$ will not overestimate the value compared to the behavior that is most familiar with that state. Thus, at each time step, if one or more behaviors are familiar with the current state, the performance of the chosen behavior will not be much worse than its value function estimate. In this manner, ROAM adjusts value estimates based on degree of familiarity, mitigating overestimation risks. The ability to adjust the $\beta$ parameter offers a flexible framework to optimize for the behavior with the highest value at a given state, which will be at least as suitable as the most familiar behavior.

In the next section, we find empirically that after fine-tuning with the additional cross-entropy loss, ROAM is able to effectively select a relevant behavior for a given state on-the-fly, leading to robust and fast adaptation to OOD situations.

## 5 EXPERIMENTAL RESULTS

In this section, we evaluate the performance of ROAM empirically and assess how effectively it can adapt on-the-fly to new situations. Concretely, we aim to answer the following questions: (1) In simulated and real-world settings, how does ROAM compare to existing methods when given diverse prior behaviors/data and deployed in novel situations? (2) How does the additional cross-entropy term in the loss function of ROAM contribute to more successful utilization of the prior behaviors? In the remainder of this section, we describe our experimental setup and present our results on both a simulated and a real-world Go1 quadrupedal robot. For qualitative video results, see our project webpage: https://sites.google.com/view/adapt-on-the-go/home.

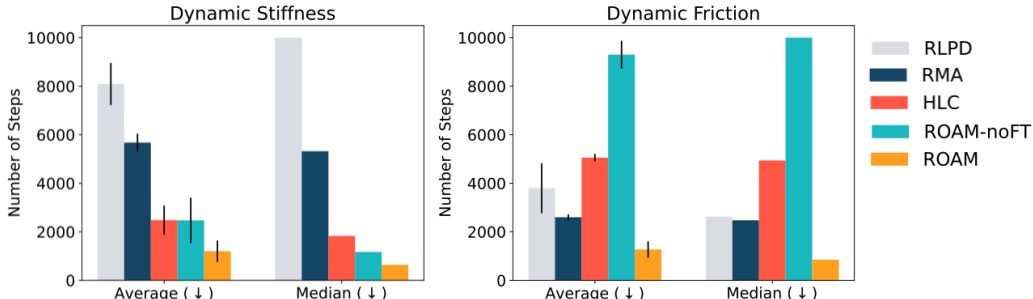

Figure 3: **Results on the simulated Go1 robot.** In both evaluation settings, ROAM is over 2x as efficient as all comparisons in terms of both average and median number of steps taken to complete the task.

**General Experimental Setup.** We use the setting of locomotion deployment to evaluate ROAM, as it is a challenging setting for adaptation, where the agent may naturally face a variety of different situations and must adapt its walking behavior on-the-fly without any additional supervision or human intervention. We use a Go1 quadruped robot from Unitree and MuJoCo (Todorov et al., 2012) for simulation. We implemented all methods on top of the same state-of-the art implementation of SAC from Smith et al. (2022) as the base learning approach, with regularization additions following DroQ (Hiraoka et al., 2021), and RLPD (Ball et al., 2023) for methods that do online fine-tuning. We include details and hyperparameters in Appendix A.5.

**Comparisons.** We evaluate ROAM along with the following prior methods: (1) RLPD Fine-tuning (Ball et al., 2023), where we fine-tune a single policy using all the data from the prior behaviors with RLPD; (2) RMA (Kumar et al., 2021), which trains a base policy and adaptation module that estimates environment info; (3) High-level Classifier (HLC), which trains a classifier on the data buffers of the pre-trained behaviors and uses it to select which behavior to use at a given state, as a representative method for those that train an additional behavior selection network, similar to work by Han et al. (2023). We additionally consider an ablation, ROAM-NoFT, which uses the values of the prior behaviors to choose among behaviors but does not fine-tune with the additional cross-entropy loss. We give RMA access to unlimited online episode rollouts in each of the prior MDPs during pre-training, while all other methods use the same set of offline data and prior behaviors that are pre-trained in the prior MDPs. All methods are evaluated in the target MDP in a single episode without human interventions, and results are averaged across 10 trials.

## 5.1 SELECTING RELEVANT BEHAVIORS IN SIMULATION

**Setup.** In our simulation experiments, we evaluate in two separate settings. The first setting simulates a situation where different joints become damaged or stuck during the robot's lifetime. It uses 9 prior behaviors: each is a different limping behavior with a different joint frozen. In the single life, the agent must walk a total distance of 10 meters, and every 100 steps, one of the 3 remaining joints *not covered in the prior data* is frozen, and the agent must adapt its walking behavior to continue walking. The second setting simulates a situation where the robot encounters different friction levels on its different feet due to variation in terrain. It uses 4 different prior behaviors, each of which is trained with one of the 4 feet having low friction. During the single life, every 50 steps, the friction of one or two of the feet is changed to be lower than in the prior behaviors. To collect the prior behaviors, we train each behavior for 250k steps (first setting) or 50k steps (second setting) in the corresponding MDP, and use the last 40k steps as the prior data. We report the average and median number of steps taken to complete the task across 10 seeds along with the standard error and success rate (out of 10) in Figure 3. The agent is given a maximum of 10k steps to complete the task, and if it does not complete the task within this time, it is considered an unsuccessful trial.

**Results.** As seen in Figure 3, ROAM outperforms all other methods in all three metrics of average and median number of steps taken to complete the task as well as overall success rate. In particular, in both settings, ROAM completes the task *more than 2 times faster*, in terms of average number of timesteps, compared to the next best method. Both RLPD fine-tuning and RMA struggle on both evaluation settings, especially the stiffness setting, demonstrating the importance of adapting in the space of behaviors rather than the space of actions for more efficient adaptation. RLPD and RMA perform better in the friction evaluation, where a single policy can still somewhat adapt to the various

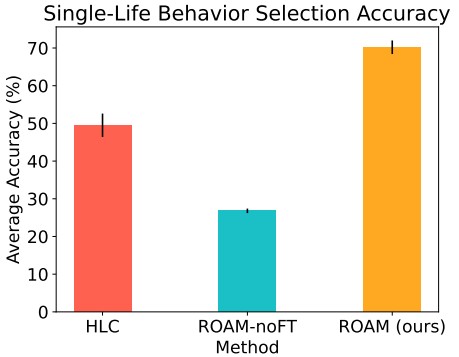

Figure 4: **Single-life Behavior Selection Accuracy.** The percent of steps where different methods select a relevant behavior for the current situation. ROAM is able to choose a relevant behavior significantly more often on average when adapting to test-time situations than HLC and ROAM-noFT.

|  | | Before FT | After FT |
|---|---|---|---|
| Changing Stiffness | Avg ID Values | 779.2 | 778.7 |
| | Avg OOD Values | 746.4 | 542.9 |
| Changing Frictions | Avg ID Values | 1243.4 | 1247.6 |
| | Avg OOD Values | 1217.6 | 1192.8 |

Figure 5: **Effect of the Cross-Entropy Loss on ID and OOD values.** The average values of the different behaviors in states visited by that behavior (ID) vs states visited by other behaviors (OOD), before and after fine-tuning with the additional cross-entropy term for ROAM. ROAM is able to maintain high values of the behavior in ID states, while decreasing the value of the behavior in OOD states, which leads to better distinction between relevant behaviors at a given state.

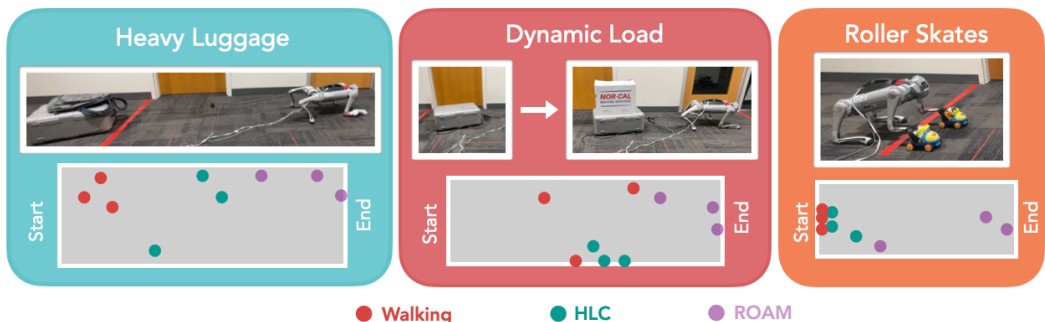

Figure 6: **Real-world single-life tasks.** We evaluate on: (1) pulling a load of heavy luggage 6.2 kg (13.6 lb), (2) pulling luggage where the weight dynamically changes between 2.36 kg (5.2 lb) and 4.2 kg (9.2 lb), and (3) moving forward with roller skates on the robot's front two feet. For each trial, for each method, we also show the locations of the robot before its first fall or readjustment (Red is Walking, Blue is HLC, and Purple is ROAM).

situations in the single life. On the other hand, HLC and ROAM-NoFT both struggle in the friction eval suite, demonstrating the importance of the additional cross-entropy term in the loss function, encouraging greater behavior specialization in different regions of the state space. These two methods perform better when the behaviors are already more distinguished, as in the limping eval suite, but they still struggle to adapt as efficiently as ROAM.

In Figures 4 and 5, we provide some additional empirical analysis of ROAM. First, in the stiffness evaluation, we plot the percent of steps where different methods select a relevant behavior during test-time deployment, where a held-out joint is frozen and a relevant behavior is one where an adjacent joint on the same leg is frozen or the same joint on an adjacent leg is frozen, and we see that ROAM is able to choose the most relevant behavior on average significantly more frequently than HLC and ROAM-noFT, which often select a behavior that is not relevant to the current situation. In addition, we record the average values of the different behaviors in states visited by that behavior (ID) vs states visited by other behaviors (OOD), before and after fine-tuning with the additional cross-entropy term for ROAM. We find that ROAM is able to effectively maintain high values of the behavior in familiar states, while decreasing the value of the behavior in unfamiliar OOD states, leading to better distinction between relevant behaviors at a given state.

## 5.2 ADAPTING ON-THE-GO1 IN THE REAL WORLD

**Setup.** On the Go1 quadruped robot, we evaluate ROAM in a setting where we have a fixed set of five prior behaviors: walking, and four different behaviors where each of the legs has a joint frozen. We pre-train a base walking behavior in 18k steps and train the other behaviors by fine-tuning the walking behavior for an additional 3k steps with one of the joints frozen, all from scratch in the real

| | Heavy Luggage | | Dynamic Luggage Load | | Roller Skates | |
|---|---|---|---|---|---|---|
| | Avg. Time (s) ↓ | Falls ↓ | Avg. Time (s) ↓ | Falls ↓ | Avg. Time (s) ↓ | Falls ↓ |
| Walking | 45.3 | 2.3 | 32 | 1 | NC | NC |
| HLC | 42.7 | 3 | 28.3 | 1.3 | 62.3 | 2.7 |
| ROAM (ours) | **25.7** | **0.7** | **24.3** | **0.3** | **27.3** | **1** |

Table 1: **Results on the real Go1 robot on 3 different tasks**: On all 3 tasks, across 3 trials for each method, ROAM significantly outperforms both comparisons in terms of both average wall clock time (s) and number of falls or readjustments needed to complete the task in a single life. NC (no complete) indicates that the task was not able to be successfully completed at all with the given method.

world using the system from Smith et al. (2022). During single-life deployment, we evaluate on the following three tasks: (1) Heavy Luggage: the robot must walk from a starting line to a finish line, while pulling a box that is 6.2 kg (13.6 lb) attached to one of the back legs. In addition, one of the front legs is one leg is covered by a low-friction plastic protective layer and the robot has to figure out how to adapt to walk with this leg. (2) Dynamic Luggage Load: the robot must walk from a starting line to a finish line, while adapting on-the-fly to a varying amount of weight between 2.36 kg (5.2 lb) and 4.2 kg (9.2 lb). We standardize each trial by adding and removing weight at the same distance from the start position. (3) Roller Skates: we fit the robot's front two feet into roller skates, and the robot must adapt to its behavior to slide its forward legs and push off its back legs in order to walk to the end line. for each of the three trials for each task. We report the average wall clock time in seconds and number of falls or readjustments needed to complete the task in a single life across 3 trials for each method. If the robot falls, a get-up controller (we use the open-sourced policy from Smith et al. (2022)) is triggered to reset the robot back to a standing position. If it walks into a wall, it is readjusted to face the correct direction. The tasks are shown in Figure 6, along with the locations of the robot before its first fall or readjustment for each method for each trial, where the red dots correspond to the Walking policy, blue to HLC, and purple to ROAM.

**Results.** Although none of the prior behaviors are trained to handle these specific test-time scenarios, the robot can leverage parts of the prior behaviors to complete the task. As shown in Table 1,ROAM significantly outperforms using a high-level classifier (HLC) as well as the baseline walking policy in terms of both average wall clock time and number of falls or readjustments at single-life time on all three real-world tasks. Qualitatively, the other methods have trouble pulling luggage consistently forward, whereas our method often chooses the behavior where a joint is frozen on the leg with the luggage attached, as this behavior uses the robot's other three legs to pull itself forward more effectively. The other methods struggle particularly on the roller skates task, which has drastically different dynamics from typical walking and especially relies on choosing relevant behaviors that heavily use the back legs. As seen in Figure 6, for all three tasks, HLC and the standard walking policy often fall or need to be readjusted very early in each single-life trial, whereas ROAM gets much closer to the finish line and often even completes the task without any falls or readjustments.

## 6 CONCLUSION AND FUTURE WORK

We introduced Robust Autonomous Modulation (ROAM), which enables agents to rapidly adapt to changing, out-of-distribution circumstances during deployment. Our contribution lies in offering a principled, efficient way for agents to leverage pre-trained behaviors when adapting on-the-fly. Through a value-based mechanism, ROAM identifies the most relevant pre-trained behaviors in real-time at each time-step without any human supervision. Our theoretical analysis confirms the effectiveness of our behavior modulation strategy, showing why a suitable behavior will be chosen for a given state with ROAM. On simulated tasks and a complex real-world tasks with a Go1 quadruped robot, we find that our method achieves over 2x efficiency in adapting to new situations compared to existing methods. While ROAM offers significant advances in enabling agents to adapt to out-of-distribution scenarios, one current limitation lies in the dependency on the range of pre-trained behaviors; some scenarios may simply be too far out-of-distribution compared to the available prior behaviors. For example, an agent trained primarily in walking tasks would struggle to adapt to the requirement of jumping over an obstacle. Future work could explore integrating ROAM into a lifelong learning framework, allowing agents to continuously expand their repertoire of behaviors, thereby increasing their adaptability to more unforeseen situations. We hope that ROAM may open new possibilities for more versatile and self-reliant autonomous systems.

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

## A  APPENDIX

### A.1  PROOFS FOR THEORETICAL ANALYSIS

In this section, we present the full proofs for Section 4.2. Following the same notation, let behavior $i$ be associated with reward $R_i = \mathbb{E}_{a \in \pi_i}[R_i(s, a)]$ and dynamics function $P_i$. Our modified Bellman operator is $\hat{B}^\pi V = \left[\hat{B}^{\pi_i} V_i\right]_{i=1}^n$, where for each value function $V_i$, the modified Bellman operator is the following:

$$(\hat{B}^{\pi_i} V_i)(s) = \left\{ (1 - \beta)\left( R_i(s) + \gamma \sum_{s' \in S, a \in A} \pi_i(a|s) P_i(s'|s, a) V_i(s') \right) + \beta \left( \frac{\exp(\tau V_i(s))}{\sum_{k=1}^n \exp(\tau V_k(s))} \right) \right\}.$$

**Lemma A.1.** *There exists $\tau > 0$, our modified Bellman operator $\hat{B}^\pi V = [\hat{B}^{\pi_i} V_i]_{i=1}^n$ is a contraction under the $\mathcal{L}^\infty$ norm.*

*Proof.* In order to show this, we first show that for any element $V_i$ of the full vector of value functions $V$, $(\hat{B}^{\pi_i} V_i)(s)$ is a contraction. For all $s \in \mathcal{S}$ and any two $V_i, V_i'$ value functions:

$$|(\hat{B}^\pi V_i)(s) - (\hat{B}^\pi V_i')(s)| = |(1 - \beta)\left( R_i(s) + \gamma \sum_{s' \in S, a \in A} \pi_i(a|s) P_i(s'|s, a) V_i(s') \right) + \beta \left( \frac{\exp(\tau V_i(s))}{\sum_{k=1}^n \exp(\tau V_k(s))} \right)$$

$$- (1 - \beta)\left( R_i(s) + \gamma \sum_{s' \in S, a \in A} \pi_i(a|s) P_i(s'|s, a) V_i'(s') \right) - \beta \left( \frac{\exp(\tau V_i'(s))}{\sum_{k=1}^n \exp(\tau V_k'(s))} \right) |$$

$$= |(1 - \beta)\left( \gamma(\sum_{s' \in S, a \in A} \pi_i(a|s) P_i(s'|s, a)(V_i(s') - V_i'(s'))\right)$$

$$+ \beta \left( \frac{\exp(\tau V_i(s))}{\sum_{k=1}^n \exp(\tau V_k(s))} - \frac{\exp(\tau V_i'(s))}{\sum_{k=1}^n \exp(\tau V_k'(s))} \right) |.$$

The first term becomes

$$|\mathbb{E}_{a \in A}(1 - \beta)\left( \gamma(\sum_{s' \in S, a \in A} \pi_i(a|s) P_i(s'|s, a)(V_i(s') - V_i'(s'))\right)| \le \gamma(1 - \beta)\max_{a \in A} \sum_{s' \in S} P_i(s'|s, a)|V_i(s') - V_i'(s'))|$$

$$\le \gamma(1 - \beta)\|V_i - V_i'\|_\infty.$$

We now focus on the second term: Since the softmax function is Lipschitz continuous with respect to the $L_2$ norm (Gao & Pavel, 2017), for all $V, V' \in \mathbb{R}$, there exists $C > 0$ such that

$$\|\left( \frac{\exp(V_i)}{\sum_{k=1}^n \exp(V_k)} - \frac{\exp(V_i')}{\sum_{k=1}^n \exp(V_k')} \right)\|_2 \le C\|V_i - V_i'\|_2 \le C\sqrt{|S|}\|V_i - V_i'\|_\infty.$$

Let $0 < \tau < 1/(C\sqrt{|S|}) < \infty$. Then

$$\|\left( \frac{\exp(\tau V_i)}{\sum_{k=1}^n \exp(\tau V_k)} - \frac{\exp(\tau V_i')}{\sum_{k=1}^n \exp(\tau V_k')} \right)\|_\infty \le C\sqrt{|S|}\|\tau V_i - \tau V_i'\|_\infty < \|V_i - V_i'\|_\infty.$$

Therefore, by the triangle inequality, for any $s \in S$,

$$\|\hat{B}^\pi V_i - \hat{B}^\pi V_i'\|_\infty = \max_s |\hat{B}^\pi V_i(s) - \hat{B}^\pi V_i'(s)|$$

$$\le \gamma(1 - \beta)\|V_i - V_i'\|_\infty + \beta\|V_i - V_i'\|_\infty$$

$$< \|V_i - V_i'\|_\infty.$$

Thus, considering our full modified Bellman operator, on $n$ value functions $V_1, ..., V_n$, we have

$$
\begin{aligned}
\|\hat{B}^\pi V - \hat{B}^\pi V'\|_\infty &= \max_i \max_s |\hat{B}^\pi V_i(s) - \hat{B}^\pi V_i'(s)| \\
&< \max_i \|V_i - V_i'\|_\infty \\
&< \|V - V'\|_\infty.
\end{aligned}
$$

$\square$

**Theorem A.2.** *Let $p_i(s)$ denote the state visitation probability for a behavior $b_i$ at state $s$. For any state $s$ that is out of distribution for behavior $b_i$ and is in distribution for another behavior $b_j$, i.e. $p_i(s) \ll p_j(s)$, if $1 > \beta > 0$ is chosen to be large enough, then the value of behavior $i$ learned by ROAM will be bounded above compared to value of behavior $b_j$, i.e. $V_i(s) \leq V_j(s)$.*

*Proof.* By Lemma A.1, for some temperature $\tau > 0$, by the contraction mapping theorem, our modified Bellman operator will lead to a fixed point. In the following, we characterize this fixed point and use it to analyze how ROAM will adjust value estimates based on degree of familiarity. Our loss function is defined based on our modified Bellman operator, optimizing for the fixed point $\hat{B}^\pi V = V$, as follows:

$$
\mathcal{L}_{\text{fine-tune}} = (1 - \beta) \sum_i \sum_{s \sim \mathcal{D}_i} p_i(s) \left[ (R_i(s) + \gamma \mathbb{E}_{s'} V_i(s') - V_i(s))^2 \right] + \beta \sum_j \sum_{s \sim \mathcal{D}_j} p_j(s) \left[ -V_j(s) + \log \sum_{k=1}^n \exp(V_k(s)) \right].
$$

Taking the derivative with respect to $V_i(s)$, we have:

$$
\begin{aligned}
\frac{\partial \mathcal{L}_{\text{fine-tune}}}{\partial V_i(s)} &= 2(1 - \beta)p_i(s)(\gamma p(s|s, a) - 1)\left(R_i(s) + \gamma \mathbb{E}_{s'} V_i(s') - V_i(s)\right) \\
&\quad + \beta \left[ p_i(s) \left( -1 + \frac{\exp(V_i(s))}{\sum_{k=1}^n \exp(V_k(s))} \right) + \sum_{j \neq i} p_j(s) \frac{\exp(V_i(s))}{\sum_{k=1}^n \exp(V_k(s))} \right] \\
&= 2(1 - \beta)p_i(s)(\gamma P_i(s|s, a) - 1)\left(R_i(s) + \gamma \mathbb{E}_{s'} V_i(s') - V_i(s)\right) + \beta \left[ -p_i(s) + \sum_j p_j(s) \frac{\exp(V_i(s))}{\sum_{k=1}^n \exp(V_k(s))} \right].
\end{aligned}
$$

Setting to 0, we have the following characterization of our fixed point, obtained as a result of Theorem A.1 for some temperature $\tau > 0$:

$$
\begin{aligned}
V_i(s) &= (R_i(s) + \gamma \mathbb{E}_{s'} V_i(s')) + \frac{\beta}{2(1 - \beta)p_i(s)(\gamma P_i(s|s, a) - 1)} \left[ -p_i(s) + \sum_j p_j(s) \frac{\exp(V_i(s))}{\sum_{k=1}^n \exp(V_k(s))} \right] \\
&= (R_i(s) + \gamma \mathbb{E}_{s'} V_i(s')) + \frac{\beta}{2(1 - \beta)(1 - \gamma P_i(s|s, a))} \left[ 1 - \sum_j \frac{p_j(s)}{p_i(s)} \frac{\exp(V_i(s))}{\sum_{k=1}^n \exp(V_k(s))} \right].
\end{aligned}
$$

We examine the behavior of $V_i(s)$ under case $p_{\text{freq}}(s) \gg p_i(s)$ for some behavior $b_{\text{freq}} \neq b_i$. Then $\frac{p_{\text{freq}}(s)}{p_i(s)}$ will dominate in the last term, so

$$
\sum_j \frac{p_j(s)}{p_i(s)} \frac{\exp(V_i(s))}{\sum_{k=1}^n \exp(V_k(s))} >> 1.
$$

Then comparing the fixed point values of $V_i(s)$ and $V_{\text{freq}}(s)$, we have

$$V_{\text{freq}}(s) - V_i(s) = (R_{\text{freq}}(s) + \gamma \mathbb{E}_{s'} V_{\text{freq}}(s')) + \frac{\beta}{2(1-\beta)(1-\gamma P_{\text{freq}}(s|s,a))}\left[1 - \sum_j \frac{p_j(s)}{p_i(s)}\frac{\exp(V_{\text{freq}}(s))}{\sum_{k=1}^n \exp(V_k(s))}\right]$$

$$- \left[(R_i(s) + \gamma \mathbb{E}_{s'} V_i(s')) + \frac{\beta}{2(1-\beta)(1-\gamma P_i(s|s,a))}\left[1 - \sum_j \frac{p_j(s)}{p_i(s)}\frac{\exp(V_i(s))}{\sum_{k=1}^n \exp(V_k(s))}\right]\right]$$

$$= ((R_{\text{freq}}(s) + \gamma \mathbb{E}_{s'} V_{\text{freq}}(s')) - (R_i(s) + \gamma \mathbb{E}_{s'} V_i(s')))$$

$$+ \frac{\beta}{1-\beta}\left(C_{\text{freq}}\left[1 - \sum_j \frac{p_j(s)}{p_i(s)}\frac{\exp(V_{\text{freq}}(s))}{\sum_{k=1}^n \exp(V_k(s))}\right] - C_i\left[1 - \sum_j \frac{p_j(s)}{p_i(s)}\frac{\exp(V_i(s))}{\sum_{k=1}^n \exp(V_k(s))}\right]\right)$$

$$> 0,$$

where $0 < C_{\text{freq}}, C_i < \infty$ constants, if

$$\frac{\beta}{1-\beta} > \frac{((R_i(s) + \gamma \mathbb{E}_{s'} V_i(s')) - (R_{\text{freq}}(s) + \gamma \mathbb{E}_{s'} V_{\text{freq}}(s')))}{\left(C_{\text{freq}}\left[1 - \sum_j \frac{p_j(s)}{p_i(s)}\frac{\exp(V_{\text{freq}}(s))}{\sum_{k=1}^n \exp(V_k(s))}\right] - C_i\left[1 - \sum_j \frac{p_j(s)}{p_i(s)}\frac{\exp(V_i(s))}{\sum_{k=1}^n \exp(V_k(s))}\right]\right)}.$$

Thus, for some $0 < \beta < 1$ large enough, $V_i(s) < V_j(s)$. □

To illustrate the effect of the cross-entropy loss, consider the following example of choosing between two behaviors $i$ and $j$ at state $s$, where the true values $V_i^{\text{true}}(s) < V_j^{\text{true}}(s)$. The optimal choice is to choose behavior $j$ and for sake of this example let us choose behavior $j$ if $V_i^{\text{ROAM}}(s) < V_j^{\text{ROAM}}(s)$. There are the following four cases: (1) $p_i(s) < p_j(s)$ and the initial estimated $V_i(s) < V_j(s)$. Then with any $\beta > 0$, the final $V_i^{\text{ROAM}}(s) < V_j^{\text{ROAM}}(s)$; (2) $p_i(s) < p_j(s)$ and the initial estimated $V_i(s) > V_j(s)$. Then by Theorem A.2, with large enough $\beta > 0$, the final $V_i^{\text{ROAM}}(s) < V_j^{\text{ROAM}}(s)$; (3) $p_i(s) > p_j(s)$ and the initial estimated $V_i(s) < V_j(s)$. Then as long as $\beta$ is chosen to be not too large, the final $V_i^{\text{ROAM}}(s) < V_j^{\text{ROAM}}(s)$. (4) $p_i(s) > p_j(s)$ and the initial estimated $V_i(s) > V_j(s)$. This is the only case where ROAM may not be adjusted to work well, but this case poses a difficult situation for any behavior selection method.

## A.2 Algorithm Summary

We summarize ROAM in Algorithms 1 and 2.

---

**Algorithm 1** ROAM Fine-Tuning

1: **Require:** $\mathcal{D}_i$, pre-trained critics $Q_i$
2: **while** not converged **do**
3:     **for all** $i$ in 1, ..., $N_{\text{behaviors}}$ **do**
4:         Sample $(s, a, s', r) \sim D_i$
5:         Update $Q_i$ according to Eq. 1
6:     **end for**
7: **end while**
8: **return** $Q_1, ..., Q_{N_{\text{behaviors}}}$

**Algorithm 2** ROAM Single-Life Deployment

1: **Require:** Test MDP $\mathcal{M}_{\text{test}}$, $\mathcal{D}_i$, policies $\pi_i$ and fine-tuned critics $Q_i$;
2: **Initialize:** online replay buffers $\mathcal{D}_{\text{online}}^i$; timestep $t = 0$
3: **while** task not complete **do**
4:     Compute values of each behavior $\{V_i(s_t)\}_1^{N_{\text{behaviors}}}$
5:     Sample behavior $b^*$ according to the distribution softmax($\exp(V_i(s_t))$).
6:     Take action $a_t \sim \pi_{b^*}(a_t|s_t)$.
7:     $\mathcal{D}_{\text{online}}^{b^*} \leftarrow \mathcal{D}_{\text{online}}^{b^*} \cup \{(s_t, a_t, r_t, s_{t+1})\}$
8:     $Q_{b^*}(s, a), \pi_{b^*} \leftarrow \text{RL}(Q_{b^*}(s, a), \pi_{b^*}, \mathcal{D}_{\text{online}}^{b^*})$
9:     Increment $t$
10: **end while**

---

## A.3 Additional Empirical Analysis

The cross-entropy term is a regularizer that creates a preference for skills that visit a given state more frequently. However, this is not the only criterion for selecting a skill; it is a regularizer. A skill with

higher value is still preferred if its visitation frequency is not too low, and ROAM does not exclusively always just select the most high-frequency behavior. We show this with the following experiment with results in Table 2. In the simulated stiffness suite, we held out most of the data from one of the buffers corresponding to one of the behaviors, leaving only 5k transitions compared to the original 40k, and evaluated the agent at test time in an environment suited for that behavior. We find that even with only 5k transitions (compared to 40k for all other behaviors), ROAM selects this less-frequent but suitable behavior the majority of the time, leading to similar overall performance.

| # Transitions | % Timesteps Chosen | Avg # Steps |
|---|---|---|
| 5k | 53.2 | 591.3 |
| 40k | 78.4 | 573.8 |

Table 2: **ROAM selects high-value behaviors even with lower visitation frequency.** We find that even with a much smaller buffer, and therefore lower visitation frequency for many states, ROAM still chooses that behavior when given situations suitable for it.

We next investigate the sensitivity of the $\beta$ hyperparameter. We ran ROAM with 4 different values (0.01, 0.1, 0.5, 0.9) of $\beta$ in each simulated suite and show the performance in Table 3. For both evaluations, 3 out of 4 of these values (all except 0.01) outperform all the other baselines.

Additionally, one benefit of ROAM is that the ability to switch between these policies at any timestep allows the agent to adapt to new and unforeseen situations, including those for which no single behavior is optimally suited. However, one hypothetical concern may be that frequent switching of behaviors may lead to suboptimal performance. In Table 3, we measure how often behaviors were switched and tried to see if frequency of behavior switches correlates with failure. We found no such correlation. Below, we show the percent of timesteps where the agent decides to switch behaviors, and more frequent switching does not correlate to a higher average number of steps needed to complete the task.

| | Dynamic Friction | | Dynamic Stiffness | |
|---|---|---|---|---|
| $\beta$ | Avg # Steps ($\downarrow$) | Frequency of Switching | Avg # Steps ($\downarrow$) | Frequency of Switching |
| 0.01 | 7610 +- 854 | 17.20% | 2698 +- 844 | 2.92% |
| 0.1 | 2082 +- 382 | 15.63% | 1331 +- 263 | 8.25% |
| 0.5 | 772 +- 179 | 11.85% | 628 +- 19 | 12.35% |
| 0.9 | 1466 +- 534 | 9.36% | 735 +- 54 | 13.36% |

Table 3: **Sensitivity of $\beta$ and frequency of behavior switching.** We find that a range of $\beta$ values give strong performance for ROAM.

## A.4 GENERAL EXPERIMENT SETUP DETAILS

As common practice in learning-based quadrupedal locomotion works, we define actions to be PD position targets for the 12 joints, and we use a control frequency of 20 Hz. Actions are centered around the nominal pose, i.e. 0 is standing. We describe the observations for the simulated and real-world experiments below.

We first detail the reward function we use to define the quadrupedal walking task. First, we have a velocity-tracking term defined as follows:

$$r_v(s, a) = 1 - |\frac{v_x - v_t}{v_t}|^{1.6}$$

where $v_t$ is the target velocity and $v_x$ is the robot's local, forward linear velocity projected onto the ground plane, i.e., $v_x = v_{\text{local}} \cdot \cos(\phi)$ where $\phi$ is the root body's pitch angle. We then have a term $r_{ori}(s, a)$ that encourages the robot to stay upright. Specifically, we calculate the cosine distance between the 3d vector perpendicular to the robot's body and the gravity vector ($[0, 0, 1]$). We then normalize the term to be between 0 and 1 via:

$$r_{ori}(s, a) = (0.5 \cdot \mathtt{dist} + 0.5)^2$$

where `dist` is the cosine distance as described above. We multiply $r_v$ and $r_{ori}$ so as to give reward for tracking velocity proportionally to how well the robot is staying upright. We then have a regularization term $r_{qpos}$ to favor solutions that are close to the robot's nominal standing pose. This regularization term is calculated as a product of a normalized term per-joint. Below, $\hat{q}^j$ represents the local rotation of joint $j$ of the nominal pose, and $q^j$ represents the robot's joint,

$$r^{\text{qpos}} = 1 - \prod_j \texttt{q distance}(\hat{q}^j, q^j)$$

where `q distance` is between 0 and 1 and decays quadratically until a threshold which is the robot's action limits. Specifically, we follow the reward structure put forth by Tunyasuvunakool et al. (2020). These terms comprise the overwhelming majority of weight in the final reward. We also include terms for avoiding undesirable behaviors like rocking or swaying that penalize any angular velocity in the root body's roll, pitch, and yaw. We also slightly penalize energy consumption and torque smoothness. To encourage a walking gait in particular, we added another regularization term to encourage diagonal shoulder and hip joints to be the same at any given time.

### A.5 Implementation Details and Hyperparameters

We implemented all methods, including ROAM, RMA, and HLC, on top of the same state-of-the art implementation of SAC from Smith et al. (2022) as the base learning approach. For all comparisons, we additionally use a high UTD ratio, dropout, and layernorm, following DroQ (Hiraoka et al., 2021), and for methods that do online fine-tuning, we use 50/50 sampling following RLPD (Ball et al., 2023). We use default hyperparameter values: a learning rate of $3 \times 10^{-4}$, an online batch size of 128, and a discount factor of 0.99. The policy and critic networks are MLPs with 2 hidden layers of 256 units each and ReLU activations. For ROAM, we tuned $\beta$ with values 0.01, 0.1, 0.5, 0.9.

**Simulated Experiments.** For the simulated experiments, the state space consists of joint positions, joint velocities, torques, IMU (roll, pitch, change in roll, change in pitch), and normalized foot forces for a total of 44 dimensions. For the position controller, we use $K_p$ and $K_d$ gains of 40 and 5, respectively, and calculate torques for linearly interpolated joint angles from current to desired at 500Hz. We define the limits of the action space to be 30% of the physical joint limits.

Table 4: Simulated Reward Function Parameter Details

| Parameter | Value |
|---|---|
| Target Velocity | 1.0 |
| Energy Penalty Weight | 0.008 |
| Qpos Penalty Weight | 10.0 |
| Smooth Torque Penalty Weight | 0.005 |
| Pitch Rate Penalty Factor | 0.6 |
| Roll Rate Penalty Factor | 0.6 |
| Joint Diagonal Penalty Weight | 0.1 |
| Joint Shoulder Penalty Weight | 0.15 |
| Smooth Change in Target Delta Yaw Steps | 5 |

For the first experimental setting, we train prior behavior policies with high stiffness (10.0) in 9 different individual joints. Specifically, we use the front right body joint, the front right knee joint, the front left body joint, the front left knee joint, the rear right body joint, the rear right knee joint, the rear left body joint, the rear left thigh joint, and the rear left knee joint. During deployment, we switch between 3 conditions every 100 steps. Condition 1 is applying stiffness 15.0 to the rear right thigh joint, condition 2 is applying stiffness 15.0 to the front left thigh joint, and condition 3 is applying stiffness 15.0 to the front right thigh joint. For this setting, we use $\beta = 0.5$ for ROAM.

For the second experimental setting, we train prior behavior policies with low foot friction (0.4) in each of the 4 feet. During deployment, we switch between 2 conditions every 50 steps. Condition 1 is applying a foot friction of 0.1 to the rear right foot and condition 2 is applying a foot friction of 0.01 to the front left foot and a foot friction of 0.1 too the rear right foot. For this setting, we use $\beta = 0.5$ for ROAM.

**Real-world Experiments.** For the real-world experiments, the state space consists of joint positions, joint velocities, torques, forward linear velocity, IMU (roll, pitch, change in roll, change in pitch), and normalized foot forces for a total of $47$ dimensions. We use an Intel T265 camera-based velocity estimator to estimate onboard linear velocity. We use $K_p$ and $K_d$ gains of $20$ and $1$, respectively, which are used in the position controller. We again use action interpolation, an action range of $35\%$ physical limits, and a $1$ step action history. We also use a second-order Butterworth low-pass filter with a high-cut value of $8$ to smooth the position targets. Finally, to reset the robot, we use the reset policy provided by Smith et al. (2022). We train $4$ prior behavior policies for the real-world experiments, each of which is trained with a frozen knee joint. Specifically, we train a policy with the front right knee joint frozen, the front left knee joint frozen, the rear right knee joint frozen, and the rear left knee joint frozen. $\beta = 0.5$ is used in all real-world experiments for ROAM.

Table 5: Real-world Reward Function Parameter Details

| Parameter | Value |
|---|---|
| Target Velocity | 1.5 |
| Energy Penalty Weight | 0.0 |
| Qpos Penalty Weight | 2.0 |
| Smooth Torque Penalty Weight | 0.005 |
| Pitch Rate Penalty Factor | 0.4 |
| Roll Rate Penalty Factor | 0.2 |
| Joint Diagonal Penalty Weight | 0.03 |
| Joint Shoulder Penalty Weight | 0.0 |
| Smooth Change in Target Delta Yaw Steps | 1 |

**HLC Details.** For HLC, we have an MLP that takes state as input and outputs which behavior to select in the given state. The MLP has 3 hidden layers of 256 units each and ReLU activations, and we train by sampling from the combined offline data from all prior behaviors. We use a batch size of 256, learning rate of $3 \times 10^{-4}$, and train for $3,000$ iterations.

**RMA Details.** For RMA training, we changed the environment dynamics between each episode and trained for a total of $2,000,000$ iterations. The standard architecture and hyperparameter choices from Kumar et al. (2021) were used along with DroQ (Hiraoka et al., 2021) as the base algorithm.

