# OpenReview forum: "Adapt On-the-Go: Behavior Modulation for Single-Life Robot Deployment"
_ICLR.cc/2024/Conference — Submitted to ICLR 2024_

### Official Review · Reviewer_YRpi · 2023-10-18

**Soundness:** 2 fair
**Presentation:** 3 good
**Contribution:** 2 fair
**Rating:** 3
**Confidence:** 3

**Summary:**

This paper proposes an approach for selecting a skill from a fixed repertoire of pre-trained behaviors. Though the general approach is mainly studied in the light of test-time distribution shift, the selection scheme is general and could potentially be applied to other problems, e.g., task selection in long-horizon tasks. At a high level, the methodology adds a regularization function to the advantages of the task repertoire to increase the tasks's value function according to the state visitation frequency (the more a policy observes that state, the higher the value is going to be). At test time, the action with the highest value is selected. Experiments in simulation and real world show the effectiveness of the approach over a set of baselines.

**Strengths:**

1. The problem this paper is trying to solve is important: skill selection in previously unseen scenarios is challenging. Using values for selection is not novel (See, for example, Chen et al., Sequential Dexterity: Chaining Dexterous Policies for Long-Horizon Manipulation), but the way the overall selection method is novel to me.
2. While the case study is focused on adaptation to distribution shifts, the approach could be generalized to other skill selection problems, e.g., long-horizon task execution.
3. The experiments show a good margin over other baselines, e.g. a naive high-level controller for skill selection trained with supervised learning.

**Weaknesses:**

The major methodological weakness in the problem formulation is the bias induced by the proposed cross-entropy term. As proven by Theoreom 4.2, the increase in the value function is proportional to the state visitation frequency. This is a problem because the high-level policy will select the low-level policy, which mostly visited a state, not necessarily the best available policy. For example, a policy that resets almost immediately will visit a low neighborhood of the initial state and, therefore, will be pushed up and, possibly, preferred to a policy with a higher value but visits the same region of the state space much more infrequently. I don't see how this can be prevented without an extra normalization term on the state visitation frequency.
Another problem, though less structural than the previous one, is that there might be multiple policies with similar values and the high-level policy switching between them at random. This could lead to suboptimal behavior and possibly lead to failures. I don't see that there is any measure preventing this in the current approach.

I am also not convinced by the experimental setup. I don't think I understood why policies trained in the real world are used for evaluation. This seems to be very interesting but orthogonal to the paper's contribution. This is a problem, in my opinion, because the policies can barely walk and keep balance, even without any pull forces (this can be seen at the beginning of the first video). This confounds the current experiments since the evaluation metrics are speed and stability. The gait and stability just can't be compared to the policies obtained via sim2real. It would be important to look into this and check whether this gap is still there after upgrading to a better policy. In addition, it is challenging to see whether a similar effect is happening in simulation without any visualization.

**Questions:**

Is there any difference in the policies on their original MDP after fine-tuning? Or, in other words, does the cross entropy have any effect on the policy performance?
Is there a way to quantify whether none of the available skills are good enough? (e.g., thresholding values)

---

> ### Author Response · Authors · 2023-11-16
>
> Thank you for your comments. We include additional empirical analyses and answers to individual concerns below and have revised the paper accordingly.
>
> > The major methodological weakness in the problem formulation is the bias induced by the proposed cross-entropy term. As proven by Theorem 4.2, the increase in the value function is proportional to the state visitation frequency. This is a problem because the high-level policy will select the low-level policy, which mostly visited a state, not necessarily the best available policy…
>
> The cross-entropy term is not the only criterion for selecting a behavior; it is a regularizer. A behavior with higher value is still preferred if its visitation frequency is not too low. In theory, there could be situations where the best behavior is not selected if it has not seen the state before, but if a particular skill has not been trained in a particular state, we have no way of knowing what its value is. So preferring not to select such skills leads to a conservative strategy. This is a common approach in RL (e.g., methods like LCB-VI and CQL in offline RL). Such methods could also be argued to prefer choosing familiar actions, but this does not stop them from being widely used in practice and having appealing theoretical properties. The more significant challenge is the overestimation of value in infrequent or out-of-distribution states, which our method aims to address, as it is the real problem that we face.
>
> In practice, the issue you describe does not seem to occur; ROAM does not exclusively always select the most high-frequency behavior. To show this, we ran an additional experiment, where in the simulated stiffness suite, we held out most of the data from one of the buffers corresponding to one of the behaviors, leaving only 5k transitions compared to the original 40k, and evaluated the agent at test time in an environment suited for that behavior. We find that even with only 5k transitions (compared to 40k for all other behaviors), ROAM still selects this less-frequent but suitable behavior the majority of the time, leading to similar overall performance.
>
> | # Transitions | % Timesteps Chosen | Avg # Steps |
> |---------------|---------------------|-------------|
> | 5k            | 53.2            | 591.3       |
> | 40k           | 78.4            | 573.8       |
>
>
> >  Another problem, though less structural than the previous one, is that there might be multiple policies with similar values and the high-level policy switching between them at random. This could lead to suboptimal behavior and possibly lead to failures. I don't see that there is any measure preventing this in the current approach.
>
> We did not find that frequent switching between behaviors causes any problems. In scenarios where multiple policies have similar values, often these policies are all adept at handling the given state. We find that in practice, the presence of multiple policies with similar values does not negatively affect performance. In fact, this can be an advantage: The ability to switch between these policies at each timestep allows the agent to adapt to new and unforeseen situations for which no single behavior is optimally suited.
>
> Empirically, we measured how often behaviors were switched and tried to see if frequency of behavior switches correlates with failure. We found no such correlation. Below, we show the percent of timesteps where the agent decides to switch behaviors, and more frequent switching does not lead to a higher average number of steps needed to complete the task.
>
> |      | Dynamic Friction       |                        | Dynamic Stiffness      |                        |
> |------|------------------------|------------------------|------------------------|------------------------|
> | beta | Avg # Steps            | Frequency of Switching | Avg # Steps            | Frequency of Switching |
> | 0.01  | 7610 +- 854            | 17.20%                 | 2698 +- 844            | 2.92%                  |
> | 0.1    | 2082 +- 382            | 15.63%                 | 1331 +- 263            | 8.25%                  |
> | 0.5   | 772 +- 179             | 11.85%                 | 628 +- 19              | 12.35%                 |
> | 0.9  | 1466 +- 534            | 9.36%                  | 735 +- 54              | 13.36%                 |

---

> > ### Author Response · Authors · 2023-11-16
> >
> > > I am also not convinced by the experimental setup. I don't think I understood why policies trained in the real world are used for evaluation. This seems to be very interesting but orthogonal to the paper's contribution. … In addition, it is challenging to see whether a similar effect is happening in simulation without any visualization.
> >
> > The choice of whether the initial policies are trained in the real world or in simulation is orthogonal to the contribution of the paper. The policies in simulation have quite strong gaits, and we have uploaded a sample rollout to our anonymous website (https://sites.google.com/view/adapt-on-the-go/home). Our experiments show that our method enhances performance both in simulation (with inherently stronger base policies) and in the real world (where base policies may appear weaker). The focus of our paper is not on sim to real transfer or about how to get the nicest gaits but instead on how to adapt. We chose to use skills trained in the real world because we found it to be convenient, efficient, and lead to decent gaits, and we do not see any reason why running with sim to real transfer would be more suitable for evaluating the central claims of the paper, which are about adaptation to new conditions.
> >
> > > Is there any difference in the policies on their original MDP after fine-tuning? Or, in other words, does the cross entropy have any effect on the policy performance?
> >
> > The cross-entropy loss only updates the critic's parameters and not the policy, so it does not have any effect on the policy performance.
> >
> > > Is there a way to quantify whether none of the available skills are good enough? (e.g., thresholding values)
> >
> > It would be interesting to explore adding an extra step where we threshold values and ask for help or train a new skill when none of the available skills are good enough. We leave this as a direction for future work.
> >
> > We hope that our response has addressed all your questions and concerns. If not, we are happy to engage in more discussion. We kindly ask you to let us know if you have any remaining concerns, and - if we have answered your questions - to reevaluate your score.

---

> > > ### Author Response · Authors · 2023-11-21
> > > **Checking in**
> > >
> > > We wanted to follow up on your review and our response. We are open to discussion if you have any additional questions or concerns, and if not, we kindly ask you to reevaluate your score and assessment of our work.

---

> > > > ### Author Response · Authors · 2023-11-22
> > > > **Following up**
> > > >
> > > > Thanks again for your review. We wanted to follow up again to make sure that your concerns are being properly addressed. Please let us know if you have additional questions. if all your concerns have been resolved, we would greatly appreciate it if you could reconsider and adjust your rating and evaluation of our work.

---

> > > ### Comment · Reviewer_YRpi · 2023-11-22
> > > **Possible misunderstanding, but issues are not addressed.**
> > >
> > > I thank the author for their response. However, there appears to be a misunderstanding.
> > >
> > > ```
> > > The cross-entropy term is not the only criterion for selecting a behavior; it is a regularizer. A behavior with higher value is still preferred if its visitation frequency is not too low. In theory, there could be situations where the best behavior is not selected if it has not seen the state before, but if a particular skill has not been trained in a particular state, we have no way of knowing what its value is. So preferring not to select such skills leads to a conservative strategy. This is a common approach in RL (e.g., methods like LCB-VI and CQL in offline RL). Such methods could also be argued to prefer choosing familiar actions, but this does not stop them from being widely used in practice and having appealing theoretical properties. The more significant challenge is the overestimation of value in infrequent or out-of-distribution states, which our method aims to address, as it is the real problem that we face.
> > > ```
> > > I did not mean that the problem is in the states you never saw before. Obviously, in such states, you can't do anything better than guessing; a low value is undoubtedly helpful. My issue is with states that are visited but with different frequencies by different policies. Assume $\pi_1$ and $\pi_2$ both visit state $s_1$, but $V_1=kV_2$, for some $k>0$, and that $P_{\pi_2} (s_1)>P_{\pi_1}(s_1) $. Given $P_{\pi_2} (s_1),P_{\pi_1}(s_1)$, it is possible to find a $\beta$ so that the proposed regularizer will increase the value of $\pi_2$, resulting in $V_2^*>V_1^*$. This results in the worst policy being constantly selected. This situation is not uncommon. For example, if all policies start from the same state, the "bad" policies will reset more and end up visiting more often the initial part of the MDP.
> > > I believe that the issue is not observed _in practice_ in the current setup. However, would this work on different tasks? If the paper's point is mainly empirical, I believe more evidence is required. A very similar argument stands for the problem I mentioned about switching behaviors. The fact that it does not seem to be a problem in this task does not mean the issue does not exist. A simple example would be finding a state where two policies with equivalent values move at opposite speeds. Constantly switching between them would let the robot stay in place.
> > >
> > > Thank you for adding the new results in the simulation, where the policy appears to be quite good. It would be nice to have a qualitative example of the other baselines as well and see how they fail and if these failures correlate with the real-world ones. I still think it is easier to train these policies in simulation and transfer them in the real world (so that the latter correlation point comes almost for free), but I agree (as I wrote in my original review) that this is orthogonal to the paper's contribution. The problem now (as mentioned by other reviewers as well) is in the small number of samples in the real world (3). With these samples, it is challenging to draw conclusions. The fact that the gait does not (visually) look stable could add quite a lot of noise in the statistics.

---

> > > > ### Author Response · Authors · 2023-11-22
> > > > **Response to latest reply from Reviewer YRpi**
> > > >
> > > > Thank you for your engagement. The first two weaknesses brought up are hypothetical issues that our method does not encounter, and would also similarly apply to many widely-used prior works in RL. For example, while there may exist a beta so that a suboptimal policy will be selected, there exists many beta for which this is not the case, and ROAM can be tuned to use one of those betas instead of a suboptimal one. Empirically, neither of the points raised seems to be a problem in practice in any of our simulated tasks (where we run 10 trials for each method) or the real-world tasks. Unfortunately, real-world experiments are very time consuming in part due to technical difficulties unrelated to the method, e.g. camera cables disconnecting, and ROAM showed a very significant improvement over other methods with the given trials. However, we will run 10 trials for each method for the final version of the paper if accepted. Additionally, these tasks were not cherry-picked in any way -- we are showing results for all of the tasks we ran our method on, so there is no reason to believe these issues would rise for other reasonable situations/tasks. Thus, while there may be hypothetical issues (as every method will have), we believe the ones you raised do not compromise the value of the ideas and results in this work.
> > > >
> > > > We kindly ask that you factor in our thorough attempt to address your concerns. Our work aims to address a very important problem (as you noted) and we are excited about the potential for our method, which is supported both theoretically and empirically, to encourage future work in this area.

---

### Official Review · Reviewer_TYha · 2023-11-01

**Soundness:** 3 good
**Presentation:** 3 good
**Contribution:** 2 fair
**Rating:** 6
**Confidence:** 3

**Summary:**

The authors propose a method for policy adaptation to different tasks. Instead of relying on a high level-controller to select the task appropriate behavior, they sample from the softmax distribution derived from the behavior policies’ value functions. Specifically, they add a regularization cross-entropy term (equation 1) to artificially raise the value function in the encountered states of a behavior policy while lowering the value of other behavior policies (section 4.1). In this way, they assert that the propensity for the value function to over-estimate out-of-distribution states is reduced. This facilitates selection of the appropriate or closest in-distribution behavior policy to the encountered state at run-time.

In section 4.2, theoretical analysis is provided. They present a modified Bellman operator and show that it is a contraction (lemma 4.1). Theorem 4.2 also asserts that - with appropriately selected hyperparameters – the value function of the in-distribution behavior policy should be lower.

Evaluation is done on a legged quadruped robot in both simulation and the real world where their proposed method outperforms baselines in data efficiency (Figure 3 for simulation) and general performance (Table 1 for real world results). They further validate that their approach selects the appropriate behavior for the current situation with high accuracy (Figure 4) and show how fine-tuning with the additional cross-entropy loss causes the gap in value functions of in-distribution versus out-of-distribution policies to become more apparent (Figure 5).

**Strengths:**

- State-of-the-art performance compared to recent baseline methods.
- Theoretical analysis included.
- Simulation and real-world experiments conducted.
- Ablation study included.
- The work is well written and clear.

**Weaknesses:**

- The approach introduces an additional hyperparameter $\beta$ that must be tuned. I am also unaware of how sensitive the approach is to this hyperparameter (whether most selected values will work well and beat baselines or whether only a small handful are appropriate).
- The approach somewhat changes the definition of the Bellman operator such that it also contains a notion of the policies propensity to have encountered a given state instead of being based solely on the expected cumulative reward. Moreover, some values of $\beta$ appear to have potentially odd behaviors. For example, selecting $\beta=1$ appears to make the Bellman operator no longer depend on the reward signal?
- There is perhaps some issues of fairness compared to RMA and HLC baselines. The authors use a state-of-the-art RLPD actor-citric method for their base learning approach. If RMA and HLC baseline methods also use actor-critic agents, were they also updated to use RLPD? If not, I would perhaps be concerned that the performance benefits reported may be in part due to RLPD instead of the author’s proposed method.
- Only a small number of real-world trials (3) are done and no confidence interval / variance is reported with the results (Table 1).

**Questions:**

Questions are in part copied from the weaknesses section:
- How sensitive is the approach is to the $\beta$ hyperparameter (whether most selected values will work well and beat baselines or whether only a small handful are appropriate)?
- Some values of $\beta$ appear to have potentially odd behaviors. For example, selecting $\beta=1$ appears to make the Bellman operator no longer depend on the reward signal. Can the authors clarify this?
- There is perhaps some issues of fairness compared to RMA and HLC baselines. The authors use a state-of-the-art RLPD actor-citric method for their base learning approach. If RMA and HLC baseline methods also use actor-critic agents, were they also updated to use RLPD? If not, I would perhaps be concerned that the performance benefits reported may be in part due to RLPD instead of the author’s proposed method.

---

> ### Author Response · Authors · 2023-11-16
>
> We thank you for your thoughtful comments and positive assessment of our work. Below we provide answers to your questions and concerns.
>
> > How sensitive is the approach to the $\beta$ hyperparameter (whether most selected values will work well and beat baselines or whether only a small handful are appropriate)?
>
> We ran ROAM with 4 different values (0.01, 0.1, 0.5, 0.9) of $\beta$ in each simulated suite and show the performance below. For both evaluations, 3 out of 4 of these values (all except 0.01) outperform all the other baselines.
>
> |      | Dynamic Friction | Dynamic Stiffness |
> |------|------------------|-------------------|
> | beta | Avg # Steps      | Avg # Steps       |
> |------|------------------|-------------------|
> | 0.01  | 7610 +- 854      | 2698 +- 844       |
> | 0.1    | 2082 +- 382      | 1331 +- 263       |
> | 0.5  | 772 +- 179       | 628 +- 19         |
> | 0.9  | 1466 +- 534      | 735 +- 54         |
>
> > Some values of $\beta$ appear to have potentially odd behaviors. For example, selecting $\beta=1$ appears to make the Bellman operator no longer depend on the reward signal. Can the authors clarify this?
>
> Thanks for bringing this up, and we apologize for any confusion. In practice, it makes sense to use ROAM only with values of $\beta < 1$, and we only use such $0 < \beta < 1$ in our experiments.
>
> > There is perhaps some issues of fairness compared to RMA and HLC baselines. The authors use a state-of-the-art RLPD actor-citric method for their base learning approach. If RMA and HLC baseline methods also use actor-critic agents, were they also updated to use RLPD? If not, I would perhaps be concerned that the performance benefits reported may be in part due to RLPD instead of the author’s proposed method.
>
> Thanks for this question, and we apologize for any confusion. We built all methods, including ROAM, RMA, and HLC on top of the same state-of-the art implementation of SAC as the base learning approach. For all comparisons, we additionally use a high UTD ratio, dropout, and layernorm, following DroQ (Hiraoka et al. 2022), and for any method that does fine-tuning, we use 50/50 sampling following RLPD (Ball et al. 2022). We have revised the paper to clarify this.
>
> Again, thank you for your thoughtful review. If you have any remaining questions, please let us know.

---

> > ### Comment · Reviewer_TYha · 2023-11-16
> >
> > Thank you for responding to my questions. However, I still have some confusion. In the appendix section it states that "For ROAM, we tuned $\beta$ with values 1, 10, 100, 1000." (page 17) and notably $\beta=1$ in real world experiments (page 18). Yet, you assert that $0<\beta<1$ for experiments in your reply. Can this be clarified?

---

> > > ### Author Response · Authors · 2023-11-16
> > >
> > > Thank you for your reply and we apologize for the confusion. In our initial code implementation, we did not include the (1-beta) term in front of the first term of the loss and simply scaled the second term with a range of beta values (which has the same effect). We ended up reporting those values in the first version. We reran ROAM with the above listed beta values, where we include the (1-beta) on the first term, which gives the same performance as the previous implementation. We ask that you may take a look at the updated pdf, where everything is now consistent. Please let us know if we can clarify this further. We thank you for your detailed attention, which has significantly improved the clarity of our paper.

---

### Official Review · Reviewer_K9rY · 2023-11-02

**Soundness:** 4 excellent
**Presentation:** 4 excellent
**Contribution:** 3 good
**Rating:** 8
**Confidence:** 5

**Summary:**

This paper addresses the problem of robot adaptation on the fly to unfamiliar scenarios and proposes a method for robust autonomous modulation (ROAM) that dynamically selects and adapts pre-trained behaviors to the situation.

**Strengths:**

+ The problem of adaptation on the fly is important for robotics.

+ The proposed approach seems novel and is well-justified to address on-the-fly robot adaptation.

+ Experiments using real robots are a strength and well demonstrate the proposed method.

+ Comparison with existing methods is clear in the related work section.

**Weaknesses:**

- Figure 1 motivates the problem using examples of facing various terrain and robot failure (e.g., damaged leg), but no experiments were performed on real robots in these scenarios.

- Showing on-the-fly adaptation across different scenarios (beyond dynamic payloads) could make the experiments more convincing, for example, in a scenario when a robot with a heavy payload suddenly steps on icy terrain.

**Questions:**

Please see the weakness section.

---

> ### Author Response · Authors · 2023-11-16
>
> We thank you for your thoughtful review and appreciate your acknowledgment of many strengths of our work. We aim to build upon our experimental validation to include more diverse and challenging conditions in future works, as you suggested, to further demonstrate the versatility and robustness of our method.

---

### Public Comment · ~Takuya_Hiraoka1 · 2023-11-11
**Reference?**

Very interesting work!

I have a minor comment on a reference citation:
pp. 18: RMA Details. For RMA training, we changed the environment dynamics between each episode and trained for a total of 2, 000, 000 iterations. The standard architecture and hyperparameter choices from Kumar et al. (2021) were used along with DroQ as the base algorithm.
-> RMA Details. For RMA training, we changed the environment dynamics between each episode and trained for a total of 2, 000, 000 iterations. The standard architecture and hyperparameter choices from Kumar et al. (2021) were used along with DroQ (Hiraoka et al., 2022) as the base algorithm. ??

(Hiraoka et al., 2022): Takuya Hiraoka, Takahisa Imagawa, Taisei Hashimoto, Takashi Onishi, and Yoshimasa Tsuruoka, Dropout Q-Functions for Doubly Efficient Reinforcement Learning, International Conference on Learning Representations, 2022

---

> ### Author Response · Authors · 2023-11-13
> **Thanks for the comment!**
>
> Thank you for pointing out this reference omission. We cited the algorithm and meant to add the reference as well. We'll add it in the next version.

---

> > ### Public Comment · ~Takuya_Hiraoka1 · 2023-11-15
> > **Reply**
> >
> > Thank you!

---

### Meta-Review · Area_Chair_y35m · 2023-12-04

**Metareview:**

This paper introduces a mechanism based on the perceived value of pre-trained behaviors to select and adapt pre-trained behaviors to the situation at hand.

**Reviewers have reported the following strengths:**
- Importance of the considered problem;
- Good empirical performance;
- Generalizability of the approach.

**Reviewers have reported the following weaknesses:**
- Structural issue in the formulation of the approach;
- Additional hyperparameter requiring non-trivial tuning;
- Lack of diversity among policies.

**Decision**

This paper has received three mixed reviews, ranging from a high grade with maximum confidence to a low one with mild confidence. Unfortunately, the review with the highest grade and maximum confidence is very short and low-quality. The review simply provides a short list of trivial strengths and weaknesses, without sufficiently delving into the analysis of the paper. Despite multiple reminders, the Reviewer did not participate in the discussion and acknowledged the rebuttal.

On the contrary, the other reviews have highlighted some issues with this paper, in terms of experimental analysis and soundness. I consider the second concern to be relevant. The Reviewer and authors seem to have had a fruitful discussion about the limitations of the method, which unfortunately have not been solved. In particular, the authors claim that some of the potential issues reported by the Reviewer have not been observed in the experiments. However, this might be due to a lack of targeted experiments where this issue could arise. Thus, I encourage the authors to provide more guarantees that the proposed method is not affected by the described potential issues in a future resubmission.

**Justification For Why Not Higher Score:**

N/A

**Justification For Why Not Lower Score:**

N/A

---

### Decision · Program_Chairs · 2024-01-16

Reject